# Interaction of Naturally Occurring Phytoplankton with the Biogeochemical Cycling of Mercury in Aquatic Environments and Its Effects on Global Hg Pollution and Public Health

**DOI:** 10.3390/microorganisms11082034

**Published:** 2023-08-08

**Authors:** Zivan Gojkovic, Samuel Simansky, Alain Sanabria, Ivana Márová, Inés Garbayo, Carlos Vílchez

**Affiliations:** 1Algae Biotechnology Group, CIDERTA, University of Huelva, 21007 Huelva, Spain; alaintxo99@gmail.com (A.S.); garbayo@dqcm.uhu.es (I.G.); bital.uhu@gmail.com (C.V.); 2Faculty of Chemistry, Brno University of Technology, Purkynova 118, 61200 Brno, Czech Republic; samuel.simansky@vut.cz (S.S.); marova@fch.vut.cz (I.M.)

**Keywords:** mercury cycling, phytoplankton, Hg toxicity, aquatic environments, aquatic environments

## Abstract

The biogeochemical cycling of mercury in aquatic environments is a complex process driven by various factors, such as ambient temperature, seasonal variations, methylating bacteria activity, dissolved oxygen levels, and Hg interaction with dissolved organic matter (DOM). As a consequence, part of the Hg contamination from anthropogenic activity that was buried in sediments is reinserted into water columns mainly in highly toxic organic Hg forms (methylmercury, dimethylmercury, etc.). This is especially prominent in the coastal shallow waters of industrial regions worldwide. The main entrance point of these highly toxic Hg forms in the aquatic food web is the naturally occurring phytoplankton. Hg availability, intake, effect on population size, cell toxicity, eventual biotransformation, and intracellular stability in phytoplankton are of the greatest importance for human health, having in mind that such Hg incorporated inside the phytoplankton cells due to biomagnification effects eventually ends up in aquatic wildlife, fish, seafood, and in the human diet. This review summarizes recent findings on the topic of organic Hg form interaction with natural phytoplankton and offers new insight into the matter with possible directions of future research for the prevention of Hg biomagnification in the scope of climate change and global pollution increase scenarios.

## 1. Introduction

Mercury (Hg) is a natural trace metal ubiquitous in the environment. Severe exposure can lead to damage to the central nervous system, causing tremors, distorted speech, kidney effects, respiratory failure, dizziness, blurred vision, hallucinations, and even death [1]. Certain studies have also documented developmental delays in children and adverse cardiovascular and immunological effects [2]. Recently, Hg has also been studied as an immunotoxin primarily in susceptible murine models, which demonstrated the immunotoxicity of inorganic mercury (IHg) in mouse models [3].

Mercury can enter the environment by natural geological processes or by anthropogenic activities [4]. Once in the atmosphere, it can be transported over long distances and later enter soils or waterbodies where it can be methylated. The organic methylated form of mercury (methylmercury or MeHg) is one of the most toxic pollutants [5], particularly due to its high affinity for proteins and, hence, causes retention within tissues. This leads to biomagnification along the entire food web from plankton to top predators. Hg bioaccumulation in fish is of special interest, particularly in places where the local population relies on fish as their main protein source [6].

The bioconcentration of Hg in phytoplankton represents one of the main entry points of Hg into the food web [7]. The concentration of MeHg in phytoplankton cells can be as high as 10^5^ times higher compared to MeHg concentrations in seawater [8]. Primary producers, such as phytoplankton, sustain ecosystems by biomass production, serving as a source of food for higher trophic chain levels [9]. Therefore, the exposure of phytoplankton to low concentrations of inorganic Hg (IHg) or MeHg can threaten the function of entire aquatic systems and, ultimately, human health via seafood consumption [10,11].

Anthropogenic activities, such as mineral processing, have increased atmospheric concentrations of Hg by at least a factor of three over the last century [12]. Due to the massive and continuous industrial use of Hg, its concentration has increased in certain areas to alarming levels, reaching concentrations up to 27 μg/L in coastal waters [13]. Anthropogenic activities emitting substantial amounts of mercury are steadily declining in Europe and North America but are increasing in Asia [4], which can further worsen the present situation.

Phytoplankton plays a crucial role in biogeochemical cycling and climate regulation. Increasing anthropogenic impacts on ecosystems have led to global warming of the Earth by approximately 0.6 °C over the past 100 years, which is an unprecedented increase compared with the past 1000 years [14]. Long-term climate change and large-scale climate fluctuations can further affect ecological processes that alter phytoplankton dynamics. The changing thermal structure of the water column may shift dominance toward small-sized algal cells and species that are able to regulate their buoyancy and tolerate more heat [15]. The smaller cell size of phytoplankton will lead to lower biomass production. 

This review aims to summarize recent findings on the topic of Hg forms and their interactions with natural phytoplankton and offers new insight into the matter, with possible directions of future research for the prevention of increasing Hg biomagnification in the scope of climate change and global pollution increase scenarios.

## 2. Mercury Forms in the Environment

In the aquatic environment, mercury behaves as a very reactive element and occurs in different forms, depending on the oxidation–reduction conditions. Elemental Hg^0^ is the only liquid metal under normal atmospheric conditions capable of converting to a vapor that is partially soluble in water under ordinary conditions of temperature and atmospheric pressure. Hg vapor has a great capacity for dispersion in the atmosphere due to its long half-life, which allows long-range transportation from terrestrial emission sources to very distant points [16,17]. The atmospheric residence time of Hg vapors is estimated to be approximately 1 year [18].

The main dissolved Hg species in aquatic environments are elemental mercury (Hg^0^), complexes of Hg^II^ with various organic and inorganic ligands, and organic Hg forms, namely, methylmercury (MeHg) and dimethylmercury (DMHg) [19]. The chemical behavior of the different chemical forms of Hg plays a critical role in the biogeochemical cycling of Hg.

Elemental Hg^0^ allows for long-range transport [1], but only 10 to 30% of the total dissolved Hg in the ocean and freshwater is present as elemental Hg^0^ [19]. Divalent Hg^II^ is the dominant form of Hg in aquatic systems and soils [20,21]. Only methyl- and dimethylmercury are naturally occurring in waters, with MeHg being the most ubiquitous and most toxic organomercury compound in freshwater and estuarine systems, while DMHg is not normally detected [19]. MeHg is bioconcentrated and biomagnified in aquatic food webs, reaching up to 80–100% of the total Hg (THg) measured in fish muscle [1]. The toxic responses in freshwater fish species in environments contaminated by Hg have been reported on a global scale [22].

## 3. Biogeochemical Cycling of Mercury and Methylmercury

Anthropogenic emissions have increased atmospheric concentrations of Hg by at least a factor of three over the last century [12]. Hg naturally occurs in different minerals, in which it remains relatively stable and does not present significant risks [23,24]. The problem comes when these minerals are used for different human activities. The extraction of these minerals results in the emission of large amounts of Hg into the environment. [23]. Based on recent findings, anthropogenic sources for mercury emissions include fossil fuel combustion, production of non-ferrous metals, iron and steel production, waste burning, production of cement, and some other industrial activities [25]. Certain sources state that 24% of anthropogenic mercury emissions are from coal combustion and thermal conversion [26]. Additionally, the evidence suggests that prior to the rapid industrialization in the last century, the utilization of Hg in precious metal mining further contributed to the inputs of Hg into the atmosphere and, thus, enlarged inputs of Hg into the ocean [12]. The total annual emissions of Hg into the atmosphere are estimated to be between 6000 and 9000 tons, mainly as elemental Hg^0^ and sometimes as divalent Hg^II^ [27]. According to recent studies, around 800 tons of atmospheric Hg is generated by natural processes, which makes up approximately 18% of the total atmospheric Hg pool [28].

The main sources of Hg inputs into open ocean regions include flow from rivers and estuaries, groundwater, releases from benthic sediments, hydrothermal vents, and direct atmospheric deposition [12]. Models and measurements suggest that the dominant source of Hg deposits to oceans is direct atmospheric deposition into surface waters, with global inputs ranging from 2800 to 5800 t over the past decade [12]. Another important source of Hg for the marine ecosystem is that of fluvial origin, which originates from industrial discharges that contaminate rivers with a wide variety of pollutants [29]. Furthermore, Hg vapors in the atmosphere may come into contact with suspended particles, creating bonds and adhering to them in such a way that leads to their deposition into sediments of the seabed. This way, Hg can later pass into the aquatic environment by effects of sea currents and the action of microorganisms [30]. These effects have caused current Hg levels to be five times higher in the atmosphere and two times higher in the oceans than natural levels [24].

In the environment, the formation of MeHg is mostly mediated by mercury-methylating bacteria, which mediate the conversion of inorganic divalent mercury (Hg^II^) into MeHg under oxygen-deficient conditions (see Figure 1) [31]. Such mediators include certain sulfate-reducing bacteria, iron-reducing bacteria, methanogens, and fermenters [1,31,32,33,34,35]. However, oxygenated ocean surface waters should not be neglected, as certain studies have demonstrated that approximately 20–40% of the MeHg measured below the surface mixed layer originates from the surface and then enters deeper ocean waters [31]. This methylation takes place mainly in the sediments, water columns, and periphyton [36].

Oxygen-deficient conditions of seafloor sediments (also called “dead zones”) that are rich in dissolved sulfates create ideal conditions for methylating sulfate-reducing bacteria [32]. The formation of such dead zones is accelerating due to anthropogenic eutrophication of multiple water bodies and global warming [31,32]. Various other environmental factors are also determining factors in the divalent Hg methylation process, such as temperature, pH, and the composition of media [16].

The data listed in Table 1 provide insights into the distribution of Hg and MeHg across various marine and freshwater environments. There are significant differences between the concentrations of MeHg and THg in open oceans and seas, such as the Atlantic Ocean (Southern Polar Front; 0.93 ± 0.69 ng/L) [41], and highly polluted rivers and estuaries contaminated by anthropogenic activities, such as Cauca River basin (Columbia; 650 ng/L) [42].

## 4. Impact of Anthropogenic Climate Change on Biogeochemical Cycling of Mercury 

The transfer of MeHg from the aquatic environment to the food chain is influenced by several environmental factors, including one of the most important, that is, the bioconcentration in the base organisms of the chain, such as microalgae [23,68]. The transfer of MeHg from a liquid medium to phytoplankton is a crucial step for subsequent bioaccumulation in higher organisms and will largely determine the bioconcentration in them (Figure 1) [68].

Wu et al. (2019) [69] analyzed several marine ecosystems in which MeHg levels in the marine environment were between 0.02 and 1.94 ng/L and analyzed concentrations at different levels of the food chain. In phytoplankton, levels between 1.7 and 410 ng/g were observed, while in the next level of the chain, in zooplankton, even higher levels of between 2.7 and 2600 ng/g were observed. The last level of measurement was that of planktivorous fish, where the concentrations increased to values between 24.1 and 3400 ng/g. They also analyzed the rates of direct transfer from the water to each of the three levels. Similar rates were observed for the two lower levels, but much lower in the case of fish. The authors concluded that the concentration in the lower levels was the most influential factor in the bioconcentration of higher organisms since, even though the direct transfer rate from the water to the fish turned out to be much lower, the concentrations that accumulate were much higher [69].

The mechanisms of intracellular accumulation are of interest for the development of bioremediation techniques since they enable the accumulation of contaminants inside the cell of the selected phytoplankton to be subsequently removed from the medium with greater ease. To accumulate toxic compounds inside the cell, phytoplankton require a specific tolerance mechanism to survive the harmful effects. The most common is the binding to intracellular ligands, mainly phytochelatins and sulfhydryl groups, forming cumulative metal complexes [70]. It has been observed that some species of microalgae have a great capacity for intake and subsequent intracellular accumulation of different heavy metals, which causes their concentration inside the cell [70]. This characteristic, which at first is negative, since it facilitates the transfer of Hg to the highest levels of the food chain, may be the key to the design of new bioremediation techniques. Other techniques include flocculation, chemical precipitation, ion exchange, and adsorption using activated carbon. While adsorption on activated carbon is a method with good selectivity for Hg ions, its current cost remains high due to the limited resources available for its production [71].

Phytoplankton species with a high capacity for bioaccumulation could also be used as bioindicators of water quality, achieving greater reliability than current techniques based on measurements of the aquatic environment, since they are more sensitive and react to environmental changes more quickly than other organisms [72].

Based on these data, it can be suggested that via anthropogenically induced Hg pollution and persistent biogeochemical cycling of Hg, global levels of Hg in the environment will rise. This can lead to even higher bioavailability of mercury species, higher bioaccumulation in higher trophic levels, and, consequently, higher public health risk. 

Generally, to limit pollution impacts, it is fundamental to develop rapid diagnostic methods for environmental hazard assessment. In this case, the managers of ecosystems may use ecotoxicology, the study of biota responses to toxic substances, which may shed light on the current level of toxic pollution in an environment and serve as an early warning tool [73]. Modern ecotoxicology techniques allow the use of genome sequencing to differentiate toxicants based on the gene expression profiles of exposed organisms and, thus, directly detect the earliest stages of the toxicological response [73]. Since mercury species enter the food chain through phytoplankton, they can be used as the key organism to assess Hg exposure in the environment [73].

In Table 2 are listed recent studies of bioaccumulation of MeHg in marine or freshwater phytoplankton. The data provided by the respective authors suggest that phytoplankton have indeed a high capacity to absorb MeHg from their environment and accumulate it in their biomass.

From Table 2, we can conclude that the most studied species for MeHg uptake and its aquatic chemistry are green algae *Chlamydomonas reinhardtii*, *Selenastrum capricornutum*, and *Chlorella* sp., and various marine diatoms. All studied species accumulate MeHg upon exposure, while the intracellular MeHg concentration mainly depends on the species and time of exposure. Reported MeHg uptakes are from 2 to 18 ng/g_DW_ for *C*. *reinhardtii*, and 27.9 to 400 ng/g_DW_ MeHg for *Chlorella* sp. Another *Chlorella* strain, *Chlorella autotrophica*, accumulated as high as 132.7 ng/g_DW_ MeHg upon 72 h of exposure to 3 nM MeHg in culture medium (where DW stands for the dry weight of microalgal biomass). Table 2 underlines the fact that phytoplankton readily accumulates MeHg, which is the basis for the problem of bioaccumulation and biomagnification in the aquatic food webs. On the other hand, this property gives the algae unique roles as bioremediating agents to be used in MeHg removal from contaminated water sources.

Biological and nutrient factors important in coastal areas, along with source water and circulation-driven changes, influence ocean dynamics linked to ocean ventilation and respiration and subsequent influences on DO [82]. A decrease in oceanic DO results in a significant increase in oxygen minimum zones in global water bodies. Oxygen minimum zones are defined as zones with less than 80 µM (2.9 mg/L) of DO, which deteriorate as potential habitats for those marine organisms that depend on continuous respiration [82].

Sea ice also plays a role in the freshwater and seawater budget of the global ocean. Global warming has induced increased heat release from the ocean that affects the atmosphere through thinner sea ice and more expansive areas of open water and reduces the planet’s ability to maintain global heat balance [82].

Consequently, a hypothesis can be formed that the increasing temperatures of oceans caused by anthropogenically induced climate change, coupled with other factors, such as the acidification of seas caused by increased atmospheric concentrations of CO_2_ and the depletion of nutrients from the surface waters, can lead to shifts in the composition of phytoplankton populations and can mean the extinction of many species of primary producers.

## 5. Mercury Bioaccumulation in Aquatic Food Chains

Once mercury enters the water system, it is converted by microorganisms into organic forms, such as methylmercury and dimethylmercury. Highly toxic organic forms of mercury with bioavailable properties are ingested by all kinds of organisms, thus being transferred through all the links in the food chain [30]. Ingested Hg persists in the body and bioaccumulates, so larger organisms tend to accumulate higher amounts of this element. This effect happens because their diet is based on an intake of a large number of smaller organisms, which have previously ingested Hg. For this reason, the consumption of large marine organisms, such as tuna or swordfish, can lead to health problems in the human population and in different animal species because they tend to accumulate greater amounts of Hg [30]. The consumption of marine organisms is the primary source of human MeHg exposure [8]. The bioconcentration of MeHg in phytoplankton and zooplankton can be as high as 10^5^ and 10^6^ times compared to MeHg concentrations in seawater, respectively [8]. Intracellular MeHg is later bound to proteins of phytoplankton cells and further bioaccumulated in marine food webs. Thus, as a primary entry point of Hg into aquatic food webs, algae play an important role in the intake and transformation of Hg species in aquatic ecosystems [83].

When MeHg enters the human body, the enterohepatic cycle is unable to expel it, so it is retained and substantially increases its half-life in the body [16]. The hydrophobic properties of MeHg allow it to pass the blood–brain barrier and even enter the placenta. MeHg interacts directly with both cellular and nuclear components, causing neurotoxic effects in the brain and nervous system, damaging the kidneys, and causing irreparable damage to fetuses [16,84].

Legislation regarding Hg limits in the environment and food varies by state and the environmental matrix considered. The Minamata Convention on mercury does not establish specific environmental limits, but it obliges to control and reduce Hg emissions and release globally [85]. The WHO (World Health Organization) raises awareness of Hg toxicity and exposure risks for the general population and gives an example of documented central nervous system damage in subjects exposed to 20 µg/m^3^ Hg in air for several years [86]. European Commission Directive 2008/105/EC (of the European Union) establishes environmental quality standards for water to protect aquatic organisms and ecosystems and limits Hg content to 20 ng/L in surface water [87]. The EU also has several additional regulations related to Hg, the most recent being European Commission Directive 2023/915/EC, establishing maximum levels of Hg at 1 mg/kg for fish [88].

The MeHg ion (CH_3_Hg^+^) has a great affinity for organic and inorganic sulfuric compounds, such as sulfides and thiols, the presence of which causes MeHg speciation, giving it hydrophobic properties and increasing its bioavailability [68,84]. For example, it has been observed that the MeHg complex with cysteine behaves as a mobile nutrient that is actively transported to the endosperm of rice grains and that the concentration of thiols can both promote and inhibit the methylation of IHg by anaerobic bacteria [68,84]. Generally, the methylation rate may be affected by a specific strain of bacteria and chemical structure and concentration of organic ligand and thiol compounds [89].

## 6. Effects of Mercury Exposure on Phytoplankton

Photosynthetic marine microorganisms (phytoplankton) carry out half of the global CO_2_ sequestration while generating half of the O_2_, which is equivalent to 1% of the global biomass of plants [90]. For this reason, they play a key role both in regulating the planet’s biogeochemical cycles (especially carbon cycles), as well as in the global ecosystem and climate change [90,91]. The great capacity of phytoplankton to fix CO_2_ can be very useful in the future, enabling the design of CO_2_ capture systems based on microalgae, as they need much less space and resources, in addition to fixing CO_2_ with an efficiency between 10 and 50 times higher than other photosynthetic organisms [91,92]. Furthermore, the possibility of the utilization of microalgae as a food source is becoming of greater interest since they do not compete with terrestrial crops for agricultural land [93].

Phytoplankton encompasses the free-floating photosynthetic microorganisms present in the top layer of natural waters, namely, eukaryotic algae and cyanobacteria [94]. By photosynthetic biomass production, microalgae influence the composition and productivity of communities of all higher organisms [94]. To perform photosynthesis, microalgae take up nutrients from their environment, including trace metals [94]. This greatly influences the biogeochemical cycling of these elements, as metals accumulated by phytoplankton will be further transferred to other microbial communities and grazers [94]. Microalgae can be affected by various pollutants present in aquatic ecosystems [95]. Heavy metals constitute important environmental pollutants because of their potent metabolic toxicity for organisms [96]. Heavy metals like mercury may accumulate in primary producers, such as microalgae, and, ultimately, be transferred to other trophic levels [22].

There is substantial evidence that exposure to both IHg and MeHg induces general toxic effects in primary producers, including a reduction in growth and photosynthesis, as well as oxidative stress [10,94,95]. In turn, these negative effects inhibit their development and reproduction by causing physiological and metabolic irregularities [97]. Fortunately, it is established that the concentrations of Hg typically found in water are far below the amounts that significantly affect the photosynthesis and growth of microalgae [98].

However, mercury is distinguished from other heavy metals due to its tendency to bioaccumulate along entire aquatic food webs [99]. Mercury has a specific interaction with sulfhydryl groups in enzymes, and coupled with oxidative stress caused by its exposure, mercury can exert toxicity at all trophic levels [99]. Once inside algal cells, Hg may bind to cytosolic ligands and be distributed into organelles. The principle of Hg toxicity is blocking functional groups of enzymes by either displacing the ions from these sites or by modifying their conformation [94].

Hg^II^ was proven to be highly toxic to the photosynthetic system of microalgae by affecting the electron transport chain, changing the photochemistry of photosystem II, and, ultimately, lowering the quantum yield of photosynthesis [95]. Moreover, excessive reactive oxygen species (ROS) caused by Hg^II^ exposure can cause detrimental effects on gene expression and, all in all, cellular damage [95]. Some studies have shown that at low concentrations, MeHg may not have a significant effect on the electron transport chain but rather affects the metabolism of organelles in the cytoplasm and, consequently, membrane integrity, while IHg directly affects plasma membrane integrity [10]. Certain studies have also found that genes involved in cell motility, nutrition, and amino acid metabolism of the alga *Chlamydomonas reindhartii* were downregulated even under environmental concentrations of Hg (10^−11^–10^−8^ M) [95]. 

The intake of metals in phytoplankton cells results from passive (diffusion and adsorption) and active uptake mechanisms (complexation of dissolved metals) and is driven by bioavailability conditioned by metal speciation and abundance [100]. Hg^II^ and MeHg are present in the environment in different forms, which fundamentally affect their bioavailability and toxicity for microalgae [10]. The impact of DOM is hard to predict, as in the previous studies, both increased and decreased Hg uptake by microalgae was detected. The key factors influencing this process were the concentration and composition of DOM, as well as microalgae species [73]. Higher Hg^II^ exposure concentrations further lead to higher cell uptake [95].

Both plants and animals have developed defense mechanisms to fight against mercury exposure, including phytoplankton [99]. Microalgae alleviate mercury toxicity by employing at least three intracellular or extracellular strategies [76,99] and by the increased production of antioxidants [101]. The first strategy is metal exclusion by reducing the metal-reactive cell surface with fewer ligands to limit metal accumulation [76,94]. The immobilization of Hg on the cell surface can significantly reduce metal toxicity. Some sources state that up to 56% of total accumulated cellular mercury can be stored in cellular debris fractions [99]. The second strategy is cellular mercury vaporization by reduction to dissolved gaseous Hg^0^, which is a less bioavailable form [76,99]. However, this strategy takes place only in some algae species and the detailed mechanism still seems to be unknown. The reduction of Hg has a very rapid onset and generally depends on the duration of the exposure [94]. The third strategy is to employ intracellular sulfur-rich complexes to sequester present Hg and, thus, to control its intracellular speciation and to allow separation into vacuoles [76,94,101]. The sequestration of mercury by the production of metal-binding thiol peptides is important to resist high plasmatic Hg concentrations and to restore the function of enzymes inactivated by Hg [99]. The primary species of such thiol-rich peptides found in phytoplankton are phytochelatins. Phytochelatins are produced as a response to the presence of various metals, like Cd, Cu, Pb, Ag, Zn, or Hg, in plants, algae, or yeast with a general structure of (γ-Glu-Cys)*_n_*-Gly (*n* = 2–11) [99]. Phytochelatins are synthesized by the enzyme phytochelatin synthase, with glutathione as the main precursor; however, the contribution of phytochelatins to metal detoxification is specific to each metal and algal species. The differences may further include the enzymatic synthesis of phytochelatins and the stoichiometry of binding to metals [99]. The precursor of phytochelatins, glutathione, is the main non-protein thiol, the pool of which is involved in metal sequestration as well as in the mitigation of oxidative damage in cells [99]. In the event of mercury exposure, the glutathione concentration in the cell is increased and phytochelatin synthesis is induced [94]. Both glutathione and phytochelatins are able to bind cytosolic Hg and, thus, minimize its nonspecific binding to physiologically important biomolecules; however, phytochelatins have a higher capacity to bind Hg species than glutathione [102]. Besides its role in the detoxification of some xenobiotics and metals, glutathione is employed in various metabolic processes, such as the transfer and storage of reduced sulfur and the control of oxidative stress.

The excretion of accumulated Hg seems to be a problematic detoxification mechanism because of the strong intracellular binding of Hg [94]. Furthermore, MeHg seems to be a poor inducer of phytochelatins [94]. 

## 7. Conclusions

Based on the current trends of anthropogenically induced climate change, increasing temperatures, the acidification of seas, and nutrient depletion in surface waters, it is possible to hypothesize that, until the year 2050, a substantial number of phytoplankton species will be highly endangered and a certain amount of primary producer biomass will be lost. Ever-increasing unfavorable conditions, such as seawater acidification and ocean temperature rise, coupled with the depletion of dissolved oxygen and limited nutrients, such as phosphorous, can lead to the extinction of the most sensitive phytoplankton species and drastically reduce the global population of phytoplankton, further decreasing global oxygen production. Meanwhile, global anthropogenic inputs of mercury species into oceans and the atmosphere will likely rise. Due to the persistent biogeochemical cycling of mercury, its bio-available levels will increase. Consequently, in future oceans, a lesser volume of phytoplankton may be exposed to even higher concentrations of dissolved organic mercury species, further enhancing bioaccumulation and subsequent biomagnification in all trophic levels that, as a final consequence, will present a high risk for human health. New insights into Hg and MeHg cycling and its interactions with naturally occurring phytoplankton will help prevent this worst-case scenario in the near future.

## Figures and Tables

**Figure 1 microorganisms-11-02034-f001:**
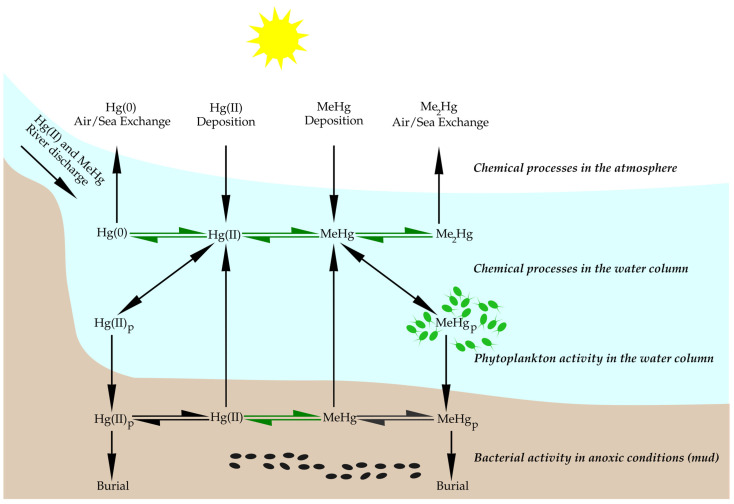
Biogeochemical cycling of Hg in coastal areas. Legend: Hg(0)—elemental mercury; Hg(II)—divalent mercury; Me_2_Hg—dimethylmercury; MeHg—methylmercury. Index p indicates that any Hg form with index p is bound to the particulate organic matter. Black arrows represent chemical processes while green arrows indicate biologically mediated processes. Green and black dots represent phytoplankton and sulfate-reducing bacteria, respectively. Sulfate-reducing bacteria thrive in environmental conditions where Hg methylation occurs with pH in the 5 to 10 range. Redox potential from slightly negative (−0.4 mV) to zero, and dissolved oxygen of less than 0.2 mg/L. Adapted from [37,38,39,40].

**Table 1 microorganisms-11-02034-t001:** Concentrations of total Hg and MeHg in different aquatic environments, according to the literature. Data are presented in the original units (ng/L, pM, and fM) that were provided by respective authors.

Location	Total Hg	MeHg	References
-	pM or ng/L	pM, fM or ng/L	-
Adour Estuary (France)	0.51–3.42 ng/L	0.025–0.081 ng L	[43]
Adriatic Sea	1.46 pM	0.28 pM	[44]
Amazon River	2.8 ng/L	-	[45]
Arctic Ocean (depth)	0.5 pM	-	[46]
Arctic Ocean (surface)	1.1 pM	-	[46]
Atlantic Ocean (Southern Polar Front)	0.93 ± 0.69 ng/L	0.26 ± 0.12 ng/L	[41]
Atlantic Ocean (north)	2.4 pM	-	[47]
Attawapiskat Drainage Basin (Canada)	0.32–7.4 ng/L	0.004–0.09 ng/L	[48]
Average in oceans	1.5 pM	-	[49]
Average in surface water of lakes and rivers	-	0.003–1.03 ng/L	[34]
Baltic Sea (northen)	1.0 ± 0.3 pM	37 ± 15 fM; 21 ± 9 fM	[37,50]
Baltic Sea (southern)	1.5 ± 0.7 pM	23 ± 13 fM	[50]
Bothnian Bay (Baltic)	1.24 ± 0.3 pM	80 ± 25 fM	[51]
Bothnian Bay (Baltic)	11.5 ± 1.66 pM	116–236 fM	[52]
Bothnian Sea (Baltic)	0.84 ± 0.24 pM	21 ± 9 fM	[51]
Carson River (Nevada)	29.1 ng/L	1.21 ng/L	[53]
Cauca River basin (Columbia)	650 ng/L	-	[42]
Crimean saline lakes	129 ng/L	-	[54]
Florida Bay (discharging canals)	3–7.4 ng/L	<0.03–52% of THg	[55]
Jiaozhou Bay (Yellow Sea)	8.46–27.3 ng/L	0.08–0.83 ng/L	[56]
Oil Sands Region Lakes (Canada)	0.4–5.3 ng/L	0.01–0.34 ng/L	[57]
Lake Titicaca (Bolivia)	-	0.01–0.18 ng/L	[58]
Lake Victoria (Africa)	3–15 ng/L	-	[59]
Mediterranean Sea	1.0 pM; 2.5 pM	-	[60,61]
Mediterranean Sea	1.46 ± 0.41 pM	0.28 ± 0.05 pM	[44]
Mekong River	1.3 ± 0.4 ng/L	0.05 ± 0.03 ng/L	[62]
Olt River (Romania)	8–88 ng/L	0.7 ng/L	[63]
Pacific Ocean	1.2 pM	-	[64]
Råne River estuary (Baltic)	2.0–5.95 pM	306 fM	[51]
South China Sea	0.8–2.3 ng/L	0,05–0.22 ng/L	[65]
Tapajos River (Brazil)	1.8 ng/L	1.46 ± 0.41 pM	[45]
In Wetlands of Rouge Park (Canada)	1.45 ± 0.91 ng/L	0.59 ± 0.45 ng/L	[66]
Yellow Sea	6.7–27.5 pM	-	[67]

**Table 2 microorganisms-11-02034-t002:** MeHg in various experimental concentrations and its accumulation by some phytoplankton species, according to the literature.

Microalgae or Cyanobacteria Strain	Experimental MeHg Conc.	Time of Exposure	MeHg Uptake	References
-	μg/L; ng/L; pM; nM	h	ng/g_DW_; µg/g_DW_; ag/µm^3^ Biomass; amol/Cell	-
*Chlamydomonas reinhardtii*	97 ± 11 pM	48 h	1.4 ± 0.19 × 10^−2^ amol/cell	[63]
*Chlamydomonas reinhardtii*	0.64−0.74 nM	48 h	17 ng/g_DW_	[74]
*Chlamydomonas reinhardtii*	5 nM	2 h	2 ng/g_DW_	[75]
*Chlamydomonas reinhardtii*	50 nM	2 h	18 ng/g_DW_	[75]
*Chlorella autotrophica*	590 ng/L	72 h	132.7 µg/g_DW_	[76]
*Chlorella* sp.	1 μg/L	72 h	27.91 µg/g_DW_	[77]
*Cyanophyceae*	0.7 ng/L	1 h	0.588 ag/µm^3^ biomass	[78]
*Isochrysis galbana*	590 ng/L	72 h	88.5 µg/g_DW_	[76]
*Isochrysis galbana*	1 μg/L	72 h	40.03 µg/g_DW_	[77]
Natural consortium: (*Oedogonium* spp. *Chlorella* spp. *Scenedesmus* spp.)	0.995 nM (200 ng/L)	6 h	340–400 ng/g_DW_	[34]
*Nitzschia closterium*	1 μg/L	72 h	32.74 µg/g_DW_	[77]
*Pelagophyceae*	0.7 ng/L	1 h	0.236 ag/µm^3^ biomass	[78]
*Schizothrix calcicola*	1.9 nM	0.083 h	356 ± 22.1 ng/g_DW_	[79]
*Selenastrum capricornutum*	1 pM (2 ng/L)	48 h	180.7 ng/g_DW_	[80]
*Selenastrum capricornutum*	233 nM	45 h	0.294 ng/g_DW_	[81]
*Synechococcus* sp.	0.7 ng/L	1 h	0.63 ag/µm^3^ biomass	[78]
*Thalassiosira pseudonana*	3 nM (600 ng/L)	72 h	22.1 µg/g_DW_	[76]
*Thalassiosira weissflogii*	1.9 nM	0.083 h	473 ± 30.5 ng/g_DW_	[79]

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
