# Peer review of "Interaction of Naturally Occurring Phytoplankton with the Biogeochemical Cycling of Mercury in Aquatic Environments and Its Effects on Global Hg Pollution and Public Health"

_microorganisms, 2023, doi:10.3390/microorganisms11082034_

Round 1

Reviewer 1 Report

This manuscript is an interesting review of mercury in aquatic environments. Please consider rethinking the last chapter, “Anthropogenic climate change and its effects on biogeochemical cycling of mercury”. The chapter deals to a minor extent with mercury circulation - please highlight this element's role in the process described.

Additional comments:

Lines 144-147, 151-152 - Is it possible to standardize mass units? 

Line 285 - double citation.

Line 287 – a typo (“;” -> ”.”).

Line 332  - the sentence require correction. 

Line 340-341 – consider adding an explanation of what “DW” is in the unit of “ng/gDW”

Table 1 requires arranging the data according to some scheme (alphabetic, geographical, quantity-based, or different).

Table 2 also requires arranging the data according to some scheme (alphabetic, geographical, quantity-based or different). Why is Nitzschia closterium highlighted? What are the letters?

Author Response

Review report from Reviewer 1

This manuscript is an interesting review of mercury in aquatic environments. Please consider rethinking the last chapter, “Anthropogenic climate change and its effects on biogeochemical cycling of mercury”. The chapter deals to a minor extent with mercury circulation - please highlight this element's role in the process described.

Answer: Thank you for very detailed and highly constructive comments that will greatly

improve the quality and scientific relevance of the manuscript.

Additional comments:

Lines 144-147, 151-152 - Is it possible to standardize mass units? 

Answer: All numerical values were converted to tons, so we only have 2 units: tons and % now.

Line 285 - double citation.

Answer: Double citation was removed (now line 566).

Line 287 – a typo (“;” -> ”.”).

Answer: Typo was corrected (now line 289).

Line 332  - the sentence require correction. 

Answer: The error was corrected.

Line 340-341 – consider adding an explanation of what “DW” is in the unit of “ng/gDW”

Answer: explanation added to this line (now lines 355-356): “(where DW stands for the dry weight of microalgal biomass)”

Table 1 requires arranging the data according to some scheme (alphabetic, geographical, quantity-based, or different).

Answer: Table 1 was rearranged in the alphabetic order of different locations names, worldwide.

Table 2 also requires arranging the data according to some scheme (alphabetic, geographical, quantity-based or different). Why is Nitzschia closterium highlighted? What are the letters?

Answer: Table 2 was rearranged in the alphabetic order of the first letter of the phytoplankton species names.

Nitzschia closterium was highlighted due to an unintended typing error, for which we apologize.

Furthermore Chapters 3 to 6 were renamed and text was rearranged according to Reviewer 2 requests.

Reviewer 2 Report

The review is somewhat disjointed, and the titles do not refer to the content of each title, and for this, rephrasing the review.

 Attention must be made to make corrections in the research

Delete the title 7. Anthropogenic climate change and its effects on biogeochemical cycling of mercury

Lines 364 to 422: These lines are just unnecessary and off-topic narration

2. Mercury forms in the environment

At the beginning of the paragraph, write a mini introduction as:

In the aquatic environment, mercury behaves as a very reactive element and occurs in different forms, depending on the oxidation-reduction conditions.

3. Mercury interference with the food chain

I prefer to change the title to Mercury bioaccumulation in aquatic food chains. What is your opinion?? And this title is the fourth.

At the beginning of the paragraph, write a mini introduction as:

Once mercury enters the water system, it is converted by microorganisms to organic forms, such as methylmercury and dimethylmercury.

4. Biogeochemical cycling of mercury and methylmercury

This title must come after the title: 2. Mercury forms in the environment

Combine titles 5 and 6 in one title.

6. Role of phytoplankton in biogeochemical cycling of mercury

In the title, you mentioned the role of the phytoplankton in biogeochemical cycling of mercury. But, in the text, you focused on the microorganisms; you know microorganisms include different kinds of organisms such as Bacteria, Fungi, Protozoa, Virus, etc. So I want to replace microorganisms with phytoplankton.

You did not mention any role of the phytoplankton, except in Table 2.

Write lines 355-363 after line 324.

7. Anthropogenic climate change and its effects on biogeochemical cycling of mercury

Delete this title and delete lines 364-422 because

I see the order of the titles is as follows:

1. Introduction

2. Mercury forms in the environment

3. Biogeochemical cycling of mercury and methylmercury

4. Impact of anthropogenic climate change on biogeochemical cycling of mercury

5.  Mercury bioaccumulation in aquatic food chains. Instead of Mercury interference with the food chain.

6. Effects of mercury exposure on phytoplankton

7. Conclusions

Author Response

Review report from Reviewer 2

The review is somewhat disjointed, and the titles do not refer to the content of each title, and for this, rephrasing the review.

Answer: Thank you for very the detailed and highly constructive comments that will greatly

improve the quality and scientific relevance of the manuscript.

Attention must be made to make corrections in the research:

Delete the title 7. Anthropogenic climate change and its effects on biogeochemical cycling of mercury

Answer: The title “7. Anthropogenic climate change and …”, was removed as suggested.

Lines 364 to 422: These lines are just unnecessary and off-topic narration

Answer: Lines 364 to 422 that followed title 7. were removed as suggested.

  1. Mercury forms in the environment

At the beginning of the paragraph, write a mini introduction as:

In the aquatic environment, mercury behaves as a very reactive element and occurs in different forms, depending on the oxidation-reduction conditions.

Answer: Mini introduction was added (lines 71– 72) at the beginning of the paragraph “2. Mercury forms in the environment”, as suggested.

  1. Mercury interference with the food chain

I prefer to change the title to Mercury bioaccumulation in aquatic food chains. What is your opinion? And this title is the fourth.

Answer: The title was changed and the order of paragraph set to 5, as suggested at the final chapter distribution below.

At the beginning of the paragraph, write a mini introduction as:

Once mercury enters the water system, it is converted by microorganisms to organic forms, such as methylmercury and dimethylmercury.

Answer: Mini introduction was added at the beginning of the paragraph (lines 439 - 440) “5. Mercury bioaccumulation in aquatic food chains”, as suggested.

  1. Biogeochemical cycling of mercury and methylmercury

This title must come after the title: 2. Mercury forms in the environment

Answer: The chapter was placed after Mercury forms in the environment chapter, as suggested

Combine titles 5 and 6 in one title.

Answer: Titles 5 and 6 were combined in one and changed to: “4. Impact of anthropogenic climate change on biogeochemical cycling of mercury”, as suggested.

  1. Role of phytoplankton in biogeochemical cycling of mercury

In the title, you mentioned the role of the phytoplankton in biogeochemical cycling of mercury. But, in the text, you focused on the microorganisms; you know microorganisms include different kinds of organisms such as Bacteria, Fungi, Protozoa, Virus, etc. So I want to replace microorganisms with phytoplankton.

Answer: In this chapter word microorganisms was replaced with phytoplankton wherever applicable.

You did not mention any role of the phytoplankton, except in Table 2.

Answer: We described the role of phytoplankton referring to it as microalgae, which as we now realize is not completely accurate and is generalization. In this chapter word microalgae was also replaced with phytoplankton wherever we talked about phytoplankton but referred to it as microalgae (which are only the part of natural aquatic phytoplankton).

Write lines 355-363 after line 324.

Answer: lines 355-363 (now lines 318-321) were placed after line 324 (now line 317)

  1. Anthropogenic climate change and its effects on biogeochemical cycling of mercury

Delete this title and delete lines 364-422 because

Answer: Title 7 was deleted and the following lines and then all chapters were ordered as suggested below.

I see the order of the titles is as follows:

  1. Introduction
  2. Mercury forms in the environment
  3. Biogeochemical cycling of mercury and methylmercury
  4. Impact of anthropogenic climate change on biogeochemical cycling of mercury
  5. Mercury bioaccumulation in aquatic food chains. Instead of Mercury interference with the food chain.
  6. Effects of mercury exposure on phytoplankton
  7. Conclusions

Answer: The work was reorganized in 7 chapters in the order that was suggested.

Round 2

Reviewer 2 Report

Please review some corrections in the text.